# Racial Disparities and Common Respiratory Infectious Diseases in Children of the United States: A Systematic Review and Meta-Analysis

**DOI:** 10.3390/diseases11010023

**Published:** 2023-01-31

**Authors:** Elizabeth A. K. Jones, Amal K. Mitra, Shelia Malone

**Affiliations:** School of Public Health, College of Health Sciences, Jackson State University, Jackson, MS 39213, USA

**Keywords:** infectious diseases, communicable diseases, contagious diseases, racial disparities, ethnic differences, racial discrimination, children

## Abstract

Due to the lack of sufficient data on the relationship between racial disparities and the occurrence of infectious respiratory diseases in children, the aim of this systematic review and meta-analysis is to evaluate the presence of racial gaps in the occurrence of respiratory infectious diseases in children. This study follows the PRISMA flow guidelines for systematic reviews and the standards of meta-analysis for 20 quantitative studies conducted from 2016 to 2022 including 2,184,407 participants. As evidenced from the review, in the U.S., racial disparities are present among children, with Hispanic and Black children carrying the burden of infectious respiratory disease occurrence. Several factors are contributory to these outcomes among Hispanic and Black children, including higher rates of poverty; higher rates of chronic conditions, such as asthma and obesity; and seeking care outside of the home. However, vaccinations can be used to reduce the risk of infection among Black and Hispanic children. Whether a child is very young or a teen, racial disparities are present in occurrence rates of infectious respiratory diseases, with the burden resting among minorities. Therefore, it is important for parents to be aware of the risk of infectious diseases and to be aware of resources, such as vaccines.

## 1. Introduction

In the United States, children have been significantly impacted by the surge in SARS-CoV-2 (COVID-19), respiratory syncytial virus (RSV), and influenza cases [1]. As result of the surge in common respiratory infectious diseases in children during the fall/winter months, especially during the current season, the consequences of racial gaps in common respiratory infectious disease occurrence may be monumental [2,3]. However, researchers have focused more on the relationship between racial gaps and infectious disease occurrence among adults [4]. There has been very little to no attention given to the relationship between racial gaps and infectious disease occurrence among children [5].

The relationship between racial gaps and infectious disease occurrence in children should be a focal point in child health research due to the impact of racial gaps on disease occurrence in vulnerable minority populations, which lead to hospitalizations, ER visits, missed days of school, burden on the healthcare system, and missed days of work for parents [6,7,8]. In the United States, 40.1 to 32.5 per 100,000 children are infected with common respiratory infections each year [9]. This statistic is likely to increase for all children, especially for vulnerable minority children, with the surge in common respiratory diseases as certain safety protocols are relaxed [10]. Due to the negative impact of racial gaps on disease infection, it is imperative to review the relationship between racial disparities and respiratory infectious disease occurrence among children in the United States in order to understand its impact of and to provide strategies that can be implemented to address respiratory infectious disease outcomes in children.

The aim of this systematic review and meta-analysis was to assess the relationship between racial gaps and the occurrence of common respiratory infectious diseases in children. In particular, the objectives of this study were: (1) to identify the impact of racial disparities and (2) to conduct quality research that can provide insight into strategies that can be implemented to address respiratory infectious disease outcomes in children.

## 2. Materials and Methods

The systematic review followed PRISMA guidelines for the inclusion of articles [11]. The study involved published quantitative studies on the relationship between racial disparities and the occurrence of infectious respiratory diseases in children aged 0–17. The infectious respiratory diseases mentioned in the study (SARS-CoV-2 (COVID-19), SARS-CoV-2 (COVID-19)-related illnesses, influenza virus, respiratory syncytial virus (RSV), human metapneumovirus (HMPV), human parainfluenza viruses (HPIVs), streptococcus, staphylococcus aureus, and rhinovirus) were selected based on the commonness of infections in children and the availability of findings within the scope of the inclusion and exclusion criteria. The inclusion and exclusion criteria applied in this review are listed in Table 1.

### 2.1. Search Guidelines

The primary search engines that were used to select articles were EBSCOhost, CINAHL, Consumer Health Complete, Health Source: Nursing/Academic, MEDLINE, Academic Search Premier, PubMed, and Google Scholar. Two of the investigators contributed to the search process. Studies were chosen for the review based on inclusion criteria, i.e., (1) articles written in English, (2) quantitative studies, (3) scholarly papers, (4) studies involving human participants between the ages of 0 and 17, (5) studies related to infectious respiratory diseases, and (6) studies related to racial disparities. The search was executed on 16 January 2023. The time limit for the studies was from 2016 to 2022. The search string is listed in Table 2.

### 2.2. Screening Guidelines

The Preferred Reporting Items for Systematic Reviews and Meta-Analysis (PRISMA) guidelines (2009) were used to direct the review process for this study [11]. Selected abstracts were reviewed to ensure that they met the eligibility criteria for inclusion. Full-text articles of eligible abstracts were retrieved and assessed based on the ability of the articles to answer the research questions and fulfill the inclusion criteria. Studies were included if a mutual consensus was reached between the two researchers.

Research information system (RIS)-formatted references were exported from the databases, and studies were automatically appraised based on the inclusion criteria, then imported into CADIMA. CADIMA is a free web tool facilitating the conduct and assuring for the documentation of systematic reviews. The 89 studies that were imported into CADIMA were evaluated based on their titles and abstracts. The researchers evaluated the studies two times before discussing whether the studies should be chosen for full-text review. Conflicts were resolved by group discussions between the researchers. After the initial discussion, the researchers agreed that 89 studies should be selected for further appraisal using the inclusion criteria. During the second phase of article assessment, the researchers again screened the 89 full-text articles two times independently. Conflicts were resolved by group discussions. After discussion, 69 articles were excluded because the they were review articles, and 20 articles were identified as appropriate for the systematic review. Owing to high heterogeneity, 10 studies were excluded, and 10 remained for meta-analysis. The PRISMA flow chart presented in Figure 1 depicts the search and inclusion process for this systematic review.

### 2.3. Quality Appraisal

Studies were appraised for quality using CADIMA. The CADIMA system was used to define standards for the critical appraisal and the rating scale of the studies. We adhered to the principles of critical appraisal tools for systematic reviews developed by the University of Adelaide, South Australia [12]. A rating scale from 0 to 4 was established for the following criteria: (1) Study design: cross-sectional, case–control, or cohort study = 1; otherwise = 0; (2) Sample size: large = 1, small = 0; (3) Use of standardized instrument(s) for data collection, such as confirmation of infection using PCR, NAAT testing, etc.= 1; not specified = 0; and (4) Selection of participants: random selection or lack of bias = 1; non-random sample, convenience sample, or presence of bias = 0. Based on the abovementioned criteria, two of the researchers evaluated each of the 20 studies independently on a scale of 0 to 4. Because no major interobserver variations or conflicts occurred during the evaluation of the quality of the studies based on the criteria listed above—study design, sample size, use of standardized instruments for data collection, and the selection of participants—an average of the two scores was calculated, as presented in Table 3 under the quality appraisal column.

## 3. Results

A summary of the methodology, characteristics of findings, presence of racial disparity, quality appraisal, and country of study is presented in Table 3. All of the reviewed studies employed a quantitative methodology [13,14,15,16,17,18,19,20,21,22,23,24,25,26,27] and used surveillance data collected from electronic health record systems. The studies were conducted from January 2016 to December 2022. In this study, we focused on the data of children only.

Fourteen studies (70%) were conducted among children between the ages of 0 and 17 years, one study [14] was conducted among children between the ages of 0 and 4 years, one study [16] was conducted among children between the ages of 0 and 5 years, one study [17] was conducted among adolescents between the ages 12 and 17, one study [25] was conducted among infants between the ages of 0 and 2 months, and two studies [24] were conducted among children and adults. However, only data reported on children within the studies [24,28] were used in this review.

The total sample size used in the reviewed studies ranged from 74 to 1,090,302, with a median sample size of 1701 (first quartile = 232 and third quartile = 17,757). The largest sample size of 1,090,302 was a result of a nationwide study that consisted of 360 counties. A total of 11 of 20 studies (55%) had sample sizes of more than 1000. All of the studies (95%) utilized standardized tools to confirm infection.

An average score of 4 of 4 (excellent) was assigned to 18 studies (90%), whereas a score of 2–3 (good) was assigned to 2 studies [19], with no studies rated as 0–1 (poor).

### 3.1. Systematic Review

#### Racial Disparities

Nineteen studies (95%) identified the presence of racial disparities in the occurrence rate of common infectious respiratory diseases, such as SARS-CoV-2 (COVID-19), SARS-CoV-2 (COVID-19)-related illnesses, influenza virus, respiratory syncytial virus (RSV), human metapneumovirus (HMPV), human parainfluenza viruses (HPIVs), streptococcus, staphylococcus aureus, and rhinovirus [13,14,15,16,17,18,19,20,21,22,23,24,26,27,28,29,30,31,32], among children in the United States. One study conducted by Perez et al. [13] identified the presence of racial disparities in the occurrence rate of infectious respiratory diseases, such as influenza, respiratory syncytial virus (RSV), human metapneumovirus (HMPV), human parainfluenza viruses (HPIVs), and SARS-CoV-2 (COVID-19), with Blacks or African Americans and Hispanics/Latinos carrying the majority (59%) of the burden of infection. Marks et al. [14] also identified the presence of racial disparities in infectious respiratory disease occurrence when observing SARS-CoV-2 (COVID-19) hospitalization rates among children aged 0–4, with Black and Hispanic children accounting for 55.5% of cases. Shi et al. [15] also identified the presence of disparity in the occurrence of respiratory infectious diseases, with Hispanic and Black children accounting for 57.6% of SARS-CoV-2 (COVID-19) hospitalizations in 14 states within the United States from July to December 2021. In a study conducted from July 2021 to January 2022, Anesi et al. [16] identified the presence of racial gaps in respiratory disease occurrence, with Black and Hispanic children constituting 52.3% of SARS-CoV-2 (COVID-19) hospitalizations in 17 states. In another study conducted by Shi et al. [17], racial gaps were observed in respiratory disease occurrence, with Black and Hispanic children accounting for 30.6% of hospitalization for SARS-CoV-2 (COVID-19) in 14 states from January 2021 to March 2021. A study by Levorson et al. [19] also established the presence of racial gaps in respiratory infectious disease occurrence, with Hispanic children having an especially high prevalence of SARS-CoV-2 (COVID-19) (26.8%). Lee et al. [19] also found a relationship between race and infectious respiratory disease occurrence, with 19.9% of Black children being hospitalized for SARS-CoV-2 (COVID-19) in New York City. This finding is extremely startling because Black children only constitute 22.2% of the New York City population. A study by Kurup et al. [20] also validated a relationship between racial disparities and disease occurrence in children, with Hispanic and Black children constituting 76.6% of positive SARS-CoV-2 (COVID-19) test results. Stierman et al. [21] also established that Black and Hispanic children constituted 53.4% of SARS-CoV-2 (COVID-19) diagnoses in 31 states. A study by Mody et al. [24] also established that Black children had a higher likelihood of testing positive for SARS-CoV-2 (COVID-19). Moreover, Graft et al. [26] established that Black and Hispanic children were more likely to test positive according to admission status (59.6%). Mannheim et al. [27] established that 78% of Black and Hispanic children tested positive for SAR-CoV-2 (COVID-19) and that Black and Hispanic children are more likely to test positive for SAR-CoV-2 (COVID-19) than non-minority racial groups. A study by Zambrano et al. [22] established that Black children have a higher likelihood of experiencing severe SARS-CoV-2 (COVID-19) outcomes. A study conducted in Mississippi by Hobbs et al. [23] revealed that Black children constituted 66% of sudden SARS-CoV-2 (COVID-19) hospitalizations from March 2020 to February 2021 at the University of Mississippi Medical Center.

Studies by Brandt et al. [28] and Artiga and Hill [32] found that Black, Hispanic, American Indian or Alaska Native, and Asian or Pacific Islander children have higher rates of SARS-CoV-2 (COVID-19) infection than White children. Studies by O’Halloran [31] and Artiga and Hill [32] also validated a relationship between infectious disease occurrence and racial gaps, finding that Black (ICU: RR:2.74, 95% CI, 2.43–3.09) (Hospitalization: RR: 2.21. 95% CI, 2.10–2.33), American Indian or Alaska Native (ICU: RR: 3.51, 95% CI, 2.45–5.05) (Hospitalization: RR: 3.00, 95% CI, 2.55–3.53), Hispanic (ICU: RR: 1.96, 95% CI, 1.73–2.23) (Hospitalization: RR: 1.87, 95% CI, 1.77–1.97), and Asian or Pacific Islander children (ICU: RR: 1.26, 95% CI, 1.16–1.38) have higher rates of ICU admission and/or hospitalization due to SAR-CoV-2 (COVID-19) infection than White children.

Finally, studies by Gualandi et al. [29] and Hansen et al. [30] further validated a relationship between racial discrimination and respiratory infectious diseases by identifying that Black children have higher rates of Methicillin-resistant Staphylococcus aureus infections [29] and cases of rhinovirus than White children [30].

However, a study conducted by Hamdan et al. [25] reported mixed findings in terms of the presence of racial disparities in the rate of occurrence of streptococcus, with no disparity present in the rate of occurrence of early-onset Group B streptococcus and racial disparity associated with the occurrence rate of late-onset Group B streptococcus, with Blacks or African Americans carrying the burden for this infection (Blacks/African Americans vs. Whites: RR, 1.55; 95% CI, 1.24–1.93; *p* < 0.001). The mixed findings of this study may be a result of newly enforced public health interventions for early-onset Group B streptococcus, such as universal screening [25].

### 3.2. Meta-Analysis

A meta-analysis of 10 studies (3090 subjects) was conducted. The data extraction included non-Hispanic Whites, non-Hispanic Blacks, and Hispanic subjects. The results relative to the included subjects were pooled and statistically analyzed to determine the effect of the relationship between race (non-Whites and Whites) and disease burden. There were significant differences in test allocation between the two groups, as noted in the forest plot in Figure 2. Although the test rates were higher in the Whites group, non-Whites had disproportionately high positive outcomes for infectious diseases compared to the non-White group. Possible limitations or moderators of this effect include study population demographics at the time of study, unknown vaccination statuses, and individual study designs.

The high *p*-value for the χ^2^ test of heterogeneity (*p* = 0.20) suggests that the heterogeneity is insignificant and that a meta-analysis is appropriate for this study, shown in Figure 3. This model assumes that all included studies measure the same thing. Several studies that measured larger population sizes were excluded in order to reduce variation across studies.

### 3.3. Risk of Bias

Figure 4 shows the authors’ judgements about each risk-of-bias item. Data are presented as percentages across all included reports or articles. There is an overall low risk of bias for this study.

### 3.4. Risk of Bias Summary

A study categorized as “unclear” risk of bias (yellow traffic light with question mark) has missing information, making it difficult to for authors to come to a consensus with respect to the assessment of limitations and potential problems. A study that is categorized as low risk of bias suggests consensus among the reviewers that the study results are valid. A study is categorized as high risk of bias when the attributes of the study design result in misleading or unclear results.

Figure 5 presents a summary report of risk of bias based on the authors’ judgements about each risk-of-bias item for each included study. Similar to Figure 3, high risk is indicated in red, yellow indicates unclear, and green indicates low risk of bias. A few items in five studies were marked as high risk of bias, whereas the majority of items or criteria were found to have low risk of bias. The investigators of the current review used some of the limitations mentioned by the authors as potential risks of bias. They are noted in Table 4 as comments about the article for the readers’ judgement. A high risk of bias is indicated in studies listed in Table 4.

## 4. Discussion

In this systematic review and meta-analysis, there was strong evidence to support the potential relationship between racial disparities and common infectious respiratory diseases in children. Higher rates of poverty among Black, Hispanic, American Indian or Alaska Native, and Pacific Islander children compared to White children seem to contribute to racial gaps in the occurrence of common infectious respiratory diseases [20,21,22,31]. Factors related to living in poverty result in increased risks of disease occurrence because Black and Hispanic children often live in multifamily dwellings, with limited financial means and no insurance [18,24,28,30,31]. These factors lead to easy transmission of infectious respiratory diseases and to difficulty in seeking treatment or paying for medicines. However, the use, awareness, and availability of free vaccines at local health departments and stores can lead to lower rates of common infectious respiratory disease in Black and Hispanic children.

Chronic diseases represent another major factor identified in this review in understanding racial differences in the occurrence of common respiratory infectious diseases. Black and Hispanic children have higher rates of asthma and obesity than non-minority groups, which contribute to increased occurrence of infectious respiratory infections among these groups of children [21,23,26,27]. Other studies have also shown links between vulnerability to respiratory infections and chronic conditions such as asthma and obesity in adults and children [34,35]. As the rate of childhood chronic conditions continues to rise, it is imperative that interventions are developed to reduce or eliminate chronic conditions among these vulnerable populations in order to reduce the occurrence of respiratory infectious diseases and severe outcomes from those infections.

Children seeking care outside of the home, such as in childcare settings or schools, is another factor contributing to racial discrimination in the occurrence of respiratory diseases [17]. Children, who are homeschooled or attend school online are less likely to contract common respiratory infectious diseases. However, 68% of children who are homeschooled or attend school online are White [36]. The underuse of homeschooling and remote learning among Blacks, Hispanics, American Indians or Alaska Natives, and Asian and Pacific Islanders may increase infection rates due to increased interactions with other children and adults in settings outside of their homes.

## 5. Limitations

This review is subject to a few potential limitations. First, although this review included multiple databases, it is possible that some studies might have been missed because of restricted inclusion criteria. Second, an inconsistency of effect or association is demonstrated by high heterogeneity prior to excluding results from three studies. Third, as noted in a study by Stierman et al. (2021) [21], the results of studies restricted to countries with complete ethnicity and race data may not be generalizable to the entire U.S. population. Further systematic reviews and meta-analyses should address studies from other countries where racial disparities are common.

## 6. Conclusions

Whether a child is very young or a teen, racial disparities are present in the occurrence rates of common infectious respiratory diseases, with most of the burden occurring among minority children, especially Blacks and Hispanics. Poverty, chronic conditions, and seeking care outside of the home are factors that could explain the racial gap in disease occurrence. These factors are associated with the racial and discriminative experiences of the parents and are a byproduct of the parents’ socioeconomic status. Unfortunately, the remnants of institutional racism and discrimination have significantly impacted not only the health of adults but also the health outcomes of their children [37]. Institutional racism has created interconnected discriminatory practices and inequities in the healthcare system, criminal justice system, educational system, and the labor and housing markets [37]. These practices have prevented equitable access to health care, equitable access to standard education or online learning options, equitable pay, and equity in homeownership and housing affordability, forcing minorities to live in multifamily dwellings [37]. Due to the racial gap, it is important that minority children, especially Black and Hispanic children, receive the care that they need to avoid infections. Therefore, it is important that minority parents use and are aware of available resources, such as vaccines. More research is needed on the relationship between racial disparities and the occurrence rates of common respiratory infectious diseases. Among many interventions, more emphasis should be placed on the implementation of public health interventions, such as universal screenings [25,38].

## Figures and Tables

**Figure 1 diseases-11-00023-f001:**
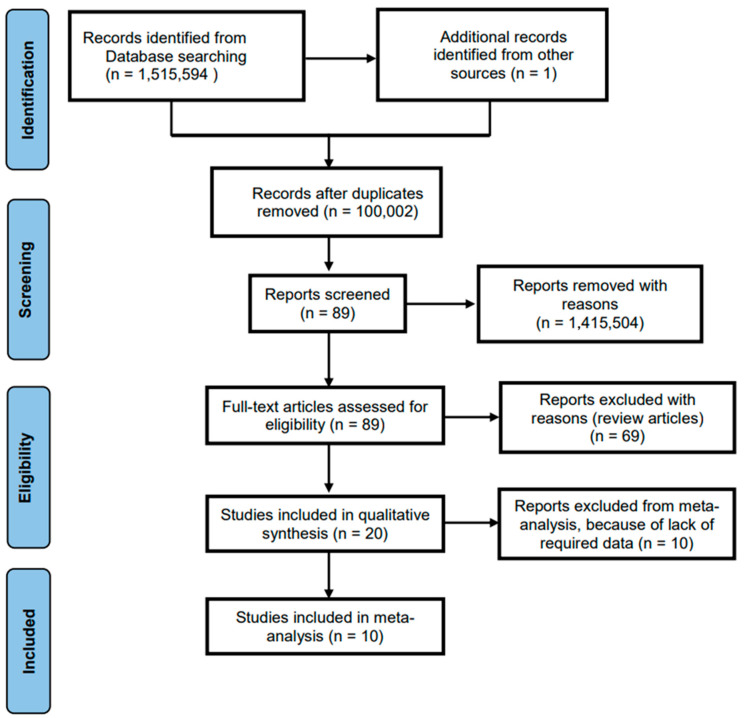
PRISMA flow chart to illustrate the article search and the inclusion process.

**Figure 2 diseases-11-00023-f002:**
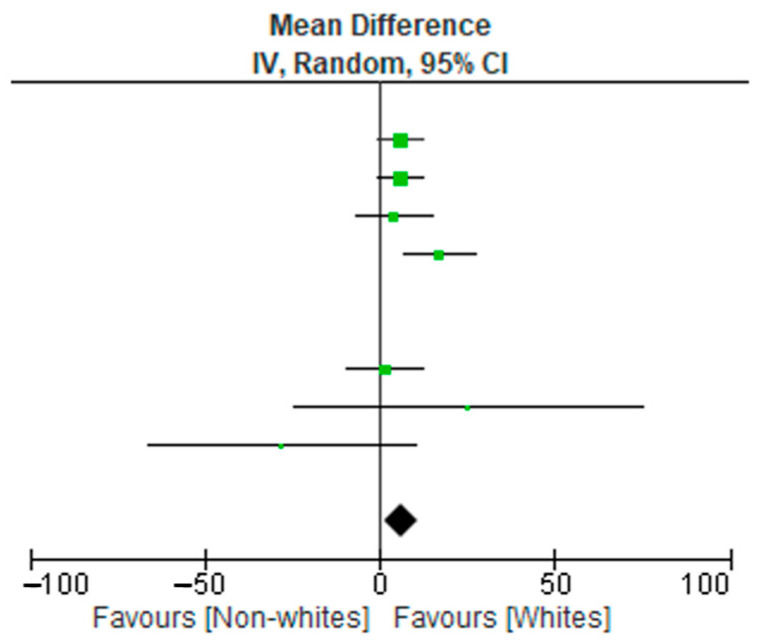
Forest plot data of 10 reviewed studies.

**Figure 3 diseases-11-00023-f003:**
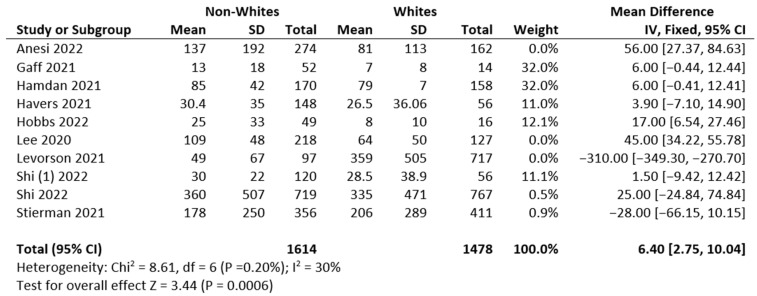
Mean, standard deviation, and heterogeneity of reviewed studies [15,16,17,18,19,21,23,25,26,33].

**Figure 4 diseases-11-00023-f004:**
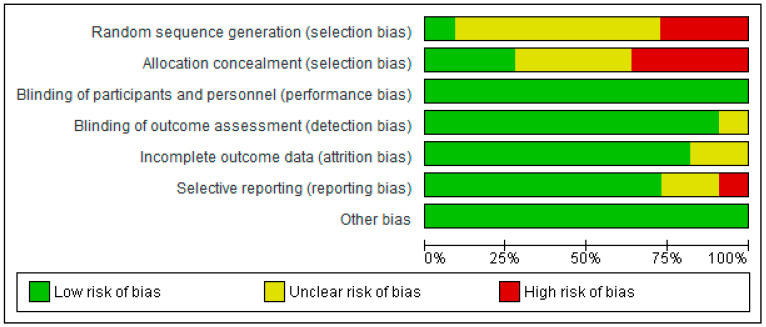
Risk of bias as assessed in 10 studies.

**Figure 5 diseases-11-00023-f005:**
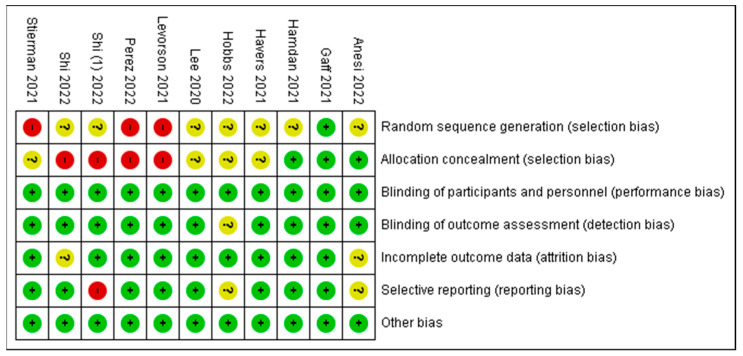
Risk-of-bias summary for 10 reviewed studies [15,16,17,18,19,21,23,25,26,33].

**Table 1 diseases-11-00023-t001:** Inclusion and exclusion criteria.

Inclusion Criteria	Exclusion Criteria
Quantitative studies	Studies published in languages other than English
Human studies	Studies that only involved participants aged 18 and older
Scholarly papers	Review articles
Age group: 0–17	Not human studies
Infectious respiratory diseases	Not in the United States
Research associated with racial disparities	

**Table 2 diseases-11-00023-t002:** Research thread for all databases.

Search Strategy	No. of Studies
Search terms used: ‘infectious diseases’ OR ‘communicable diseases’ OR ‘contagious diseases’ ‘ethnic differences’ OR ‘racial discrimination’ OR ‘children’ OR ‘kids’	1,515,595
Total number of studies excluded based on eligibility criteria	1,515,506
Total number of studies excluded because either they were review articles or they did not provide full articles	69
Total number of studies accepted and reviewed	20

**Table 3 diseases-11-00023-t003:** Relationship between racial disparities and occurrence of infectious respiratory diseases in children.

Author [Ref]	Country of Study	Type of Study	Major Findings	Racial Disparities Present	Quality Appraisal (Out of 4)
Perez et al., 2022 [13]	United States	Cohort	n = 51,441; 16,582 (32.3%) Black or African American children and 13,771 (26.8%) Hispanic children entered clinical settings with acute respiratory viruses, such as the influenza virus, respiratory syncytial virus, human metapneumovirus (HMPV), human parainfluenza (HPIVs), or SARS-CoV-2 (COVID-19), from 2016–2021.	Yes	4 (Excellent)
Marks et al., 2022 [14]	United States	Cohort	n = 2562; 719 (26.7%) Black or African Americans, 710 (28.8%) Hispanics, and 154 Asian Americans (6.0%) between the ages of 0 and 4 were hospitalized for SARS-CoV-2 (COVID-19) during the periods dominated by the pre-delta, delta, and omicron variants in 14 states from March 2020 to February 2022.	Yes	4 (Excellent)
Shi et al., 2022 [15]	United States	Cohort	n = 2100; 463 (21.8%) Hispanics and 736 (35.8%) Blacks or African Americans were hospitalized for SARS-CoV-2 (COVID-19) in 14 states from 1 July to 31 December 2021.	Yes	4 (Excellent)
Anesi et al., 2022 [16]	United States	Cohort	n = 176; 60 (34.1%) Hispanics and 32 (18.2%) Blacks or African Americans <6 months were hospitalized with SARS-CoV-2 (COVID-19) in 17 states from July 2021 to January 2022.	Yes	4 (Excellent)
Shi et al., 2022 [17]	United States	Cohort	n = 204; 117 Blacks or African Americans and 115 (30.6%) Hispanics were hospitalized for SARS-CoV-2 (COVID-19) in 14 states from January 2021 to March 2021.	Yes	4 (Excellent)
Levorson et al., 2021 [18]	United States	Cross-sectional	n = 207 (Hispanics); Hispanic children (26.8%, 55/207) had an especially high prevalence rate of SARS-CoV-2 (COVID-19).	Yes	4 (Excellent)
Lee et al., 2020 [19]	United States	Cohort	n = 223; Black or African American children constitute 22.2% of the NYC population. However, 19.9% of COVID-19 hospitalizations occurred among Black patients younger than 20; 75/223 (34%) patients in this study with COVID-19 associated multisystem inflammatory syndrome were Black.	Yes	3 (Good)
Kurup et al., 2022 [20]	United States	Cross-sectional	n = 1000; of 207 positive cases of COVID-19, 46.6% were Hispanic and 30.0% were Black.	Yes	4 (Excellent)
Stierman et al., 2021 [21]	United States	Case–control	n = 1,090,302; 473,785 (43.5%) Hispanics and 107,470 (9.9%) Blacks or African Americans were diagnosed with COVID-19 in 31 states.	Yes	4 (Excellent)
Zambrano et al., 2022 [22]	United States	Case–control	n = 241; Black or African American children had a higher likelihood of severe COVID-19 outcomes.	Yes	4 (Excellent)
Hobbs et al., 2022 [23]	United States	Cohort	n = 74; 49 (66%) Blacks or African Americans were suddenly hospitalized for COVID-19 from March 2020 to February 2021 at the University of Mississippi Medical Center in Jackson, MS.	Yes	4 (Excellent)
Mody et al., 2020 [24]	United States	Cohort	n = 934,929; Blacks had a higher likelihood of testing positive for COVID-19.	Yes	4 (Excellent)
Hamdan et al., 2021 [25]	United States	Cohort	n = 356; Whites had a higher rate of early-onset Group B *streptococcus* disease; Blacks had a higher rate of late-onset Group B *streptococcus*.	Yes/No	4 (Excellent)
Graff et al., 2021 [26]	United States	Cohort	n = 454; 23 Blacks or African Americans and 248 Hispanics tested positive for SARS-CoV-2 (COVID-19) according to admission status.	Yes	4 (Excellent)
Mannheim et al., 2021 [27]	United States	Case–control	n = 1, 302; 324 (25%) Blacks or African Americans and 695 (53%) Hispanics/Latinos tested positive for COVID-19; both Blacks or African Americans and Hispanics/ Latinos have a higher likelihood of being infected with COVID-19 than non-minority racial groups.	Yes	4 (Excellent)
Brandt et al., 2021 [28]	United States	Cohort	n = 295,642; n = 19,408 (Under 18); 6.04% of children within the study population tested positive for COVID-19. Black and Hispanic children accounted for 60–70% of COVID-19 cases among children younger than 18 years of age.	Yes	4 (Excellent)
Gualandi et al., 2018 [29]	United States	Cohort	n = 45,550; Black children had higher rates methicillin-resistant Staphylococcus aureus infections than White children.	Yes	4 (Excellent)
Hansen et al., 2022 [30]	United States	Cross-sectional	n = 16,106; Black children under 12 years of age had signigicantly higher rates of rhinovirus than non-minority racial groups.	Yes	3 (Good)
O’Halloran et al., 2021 [31]	United States	Cross-sectional	n = 15,114; Black, American Indian or Alaska Native, and Asian or Pacific Islanders had higher rates of ICU admission and hospitalization from influenza than non-minority racial groups.	Yes	4 (Excellent)
Artiga and Hill, 2021 [32]	United States	Cohort	n = 2658; American Indian or Alaska Native, Native Hawaiian and other Pacific Islander, and Hispanic children experienced the highest rates of COVID-19 infection. American Indian or Alaska Native, Hispanic, Native Hawaiian and other Pacific Islander, and Black children were two or three times more likely to be hospitalized due to a COVID-19 infection than White children; death rates among American Indian or Alaska Native and Black children were 3.5 or 2.7 times higher than among White children.	Yes	4 (Excellent)

**Table 4 diseases-11-00023-t004:** Comments on risk of bias.

Authors/Studies	Comments
Anesi et al. 2022 [16]	Self-reported data for a few participants were included in this analysis, a few infants may have been misclassified due to maternal vaccination status, or the mothers’ recollection of complete COVID-19 vaccination may be imperfect, as reported by the author as limitations. The small sample sizes mentioned by the authors resulted in wide confidence intervals.
Levorson et al. 2021 [18]	The regional population representativeness may have been affected by selection bias because the methodology focused on self-referral, and subjects in the study had blood labs completed for other clinical purposes. The authors made adjustments for these factors.
Shi et al. 2022 [15]	As mentioned by the authors, COVID-19 hospitalizations may have been overlooked because of testing practices and availability. Detailed clinical data are limited for the period during which the omicron variant was predominant (19–31 December 2021). The data do not include the peak of hospitalizations during this period; the delta variant was prevalent in late December.
Perez et al. 2022 [13]	New Vaccine Surveillance Network data are limited to enrolled consenting participants, who may not be representative of all children receiving health care, as mentioned by the authors as limitations.
Stierman et al. 2021 [21]	The proportion of positive SARS-CoV-2 test results recorded in drive-through clinics increased during the COVID-19 pandemic, as reported in the paper. This study focused on counties with complete ethnicity and race data. The results may not be generalizable to larger populations, as suggested by the authors.

## Data Availability

Not applicable.

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
