# Peer review of "Racial Disparities and Common Respiratory Infectious Diseases in Children of the United States: A Systematic Review and Meta-Analysis"

_diseases, 2023, doi:10.3390/diseases11010023_

Round 1

Reviewer 1 Report

The manuscript by Amal K. Mitra and colleagues decribes a systematic review about racial disparities and respiratory infectious diseases occurrence in children infectious disease occurrence in children of the United States. The authors urge parents to be aware of the risk of infectious diseases and available resources, such as vaccines. The manuscript is worth of being published in this journal. Minor revisions are needed.

1.      The title “Racial Disparities and Common Respiratory Infectious Diseases in Children: A Systematic Review and Meta-Analysis” could be more suitable if it’s changed to “Racial Disparities and Common Respiratory Infectious Diseases in Children of the United States: A Systematic Review and Meta-Analysis”. Because all of statistics in this review are from reported data about children of United States.

2.      The page 3, About CADIMA, no full name, only the abbreviation. The author should explain each of  abbreviations the first time it appears in the manuscript.

Author Response

Reviewer 1

Comments

Response

Lines

The title “Racial Disparities and Common Respiratory Infectious Diseases in Children: A Systematic Review and Meta- Analysis” could be more suitable if it’s changed to “Racial Disparities and Common Respiratory Infectious Diseases in Children of the United States: A Systematic Review and Meta- Analysis”. Because all of statistics in this review are from reported data about children of

United States.

The titled revised as suggested.

Line 2

The page 3, About CADIMA, no full name, only the abbreviation. The author should explain each of abbreviations

the first time it appears in the manuscript.

CADIMA is not an abbreviation. It is the full name. I added a qualifying statement about CADIMA.

Page 3

Thanks for your feedback!

Reviewer 2 Report

The manuscript is a systematic review of the racial disparities related to respiratory diseases.

Some remarks:

1- Line 55: it seems that the text is wrong;

2- Line 83 and above: there is no need to specify who performed some part of the investigation;

3- Figure 1 is bad quality;

4- Variables must be in italics;

5- Line 99: Ade-lide?

6- The text in line 147 is the label of the table;

7- I am not sure if a study with such a small number of papers is relevant;

8- Figure 2 is a table in the left part. The authors must split the figure (forest plot) and the table. In addition, table 2 was not mentioned in the text;

9- Line 204: observe the formatting;

10- The authors must describe briefly the findings of each study cited beyond the summarising found in table 3;

In my opinion, the text was written based on just a few studies. It is difficult to state that the results are statistically significant. It is necessary to increase the number of investigations, including papers from other countries. Also, the formatting must be observed. There are many errors.

Author Response

Reviewer 2

Comments

Response

Lines

1- Line 55: it seems that the text is wrong;

The text has been revised

Line 56-57

2- Line 83 and above: there is no need to specify who performed some part of the investigation;

Deleted them.

Lines 96-104

3- Figure 1 is bad quality;

Revised Figure 1

Line 96-98

4- Variables must be in italics;

Changed

Lines 112-115

5- Line 99: Ade-lide?

Sorry, the word was hyphenated, because the hyphenation option was on. The option of hyphenation has

been turned off now.

Line 112

6- The text in line 147 is the label of the table;

Changed

Line 251

7- I am not sure if a study with such a small number of papers is relevant;

I added the PubMed database to the search process. Based on the inclusion/exclusion criteria, 115 additional articles were screened with additional 5 articles being added to the study, making it a total of 20.

Lines 251-253 (Please refer to table 3)

8- Figure 2 is a table in the left part. The authors must split the figure (forest plot) and the table. In addition, table 2 was not mentioned in the text;

Changed

Lines 254-285

9- Line 204: observe the formatting;

Changed

Lines 333-335

10- The authors must describe briefly the findings of each study

Added descriptions of each study.

Lines 186-249

cited beyond the summarizing found in table 3;

In my opinion, the text was written based on just a few studies. It is difficult to state that the results are statistically significant. It is necessary to increase the number of investigations, including papers from other countries. Also, the formatting must be observed.

There are many errors.

I have added an additional 5 studies after screening and considering the inclusion/exclusion criteria.

Based on comments from reviewer 1, I changed the title of the study to “Racial Disparities and Common Respiratory Infectious Diseases in Children of the United States: A Systematic Review

and Meta-Analysis”.

Line 2-4;Lines 251-253 (Please refer to table 3); Lines 186-249

Thanks for your feedback!

Reviewer 3 Report

The authors address a topic, that is fairly covered in the literature -if one browses Racial Disparities and Common Respiratory Infectious Diseases in Children” Google gives 5,000,000 results in 0.55 seconds - however, the short paper present it from a different angle, that of a systematic review and meta-data analysis, and the paper is reasonably well structured.

There are some minor points that the authors should, however, enhance in order to better present the arguments and the findings, as indicated in the following points.

·       In the abstract specify that the value of 2,085,585 refers to the combined number of participants to the 15 studies; double check the value, which does not seem to correspond to the total on the “n values” in table 3, line 147.

·       Table 2, line 72, the numbers do not add up, if 208 (1,515,479 – 1,515,271) is the number of studies that were evaluated (line 82), by subtracting 195, the “Total number of studies excluded because either they were review articles or they did not provide full articles” the result is not 15 but 13. Based on that, adjust lines 86-90 and double-check the numbers in figure 1

·       Line 99, Ade-laide (probably it is Adelaide)

·       Line 105 rephrase and be consistent in the use of the researchers’ names

·       It is not clear if table 3 indicated in line 108 is the same as table 3 in line 147, reading “Relationship between Racial Disparities and Occurrence of Infectious Respiratory Diseases in Children”.

·       It is likewise not clear what the two scores indicated in line 108 are

·       Line 112, given the criteria of inclusion/exclusion is it necessary to reiterate that the 15 studies were all based on USA cases? The same applies to line 135

·       Line 120, based on the exclusion’s criteria it would be useful if the authors clarify why one of the studies selected refers to children and adults

·       Figure 2 is not properly explained, and in any case, it would be more appropriate if moved to line 161

·       Section 3.4 could be revisited better organising the content, moving figure 3 down, after a brief explanation of what is the meaning of the risk of bias, Concurrently the comments related to the different authors, lines 180-205, should be presented in a better and more organic form, in relation to the section (no bold style for comment lines 204-205)

·       Conclusion could be expanded giving more attention to the fact that children suffer racial disparities as a result of the racial disparities and overall discrimination suffered by their parents hence the racial gap in disease is a “by-product” of the parents’ racial socio-economic disparities    

Author Response

Reviewer 3

Comments

Response

Lines

In the abstract specify that the value of 2,085,585 refers to the combined number of participants to the 15 studies; double check the value, which does not seem to correspond to the total on the “n values” in table 3, line 147.

The number has been revised based on the total of 20 studies (instead of 15).

Line 13

Table 2, line 72, the numbers do not add up, if 208 (1,515,479 – 1,515,271) is the number of studies that were evaluated (line 82), by subtracting 195, the “Total number of studies excluded because either they were review articles or they did not provide full articles” the result is not 15 but 13. Based on that, adjust lines 86-90 and double- check the numbers in figure 1

Changed

Line 76

Line 99, Ade-laide (probably it is Adelaide)

Changed

Line 112

Line 105 rephrase and be consistent in the use of the researchers’ names

Changed- names of researchers were removed based on comments from

reviewer 2

Lines 96-104

It is not clear if table 3 indicated in line 108 is the same as table 3 in line 147, reading “Relationship between Racial Disparities and Occurrence of

Infectious Respiratory Diseases in Children”.

Changed

Line 120-123

It is likewise not clear what the two scores indicated in line 108 are..

Changed

Lines 120-123

Line 112, given the criteria of inclusion/exclusion is it necessary to reiterate that the 15 studies were all based on USA cases? The same applies

to line 135

Change Changed

Line 129

Lines 191-192

Line 120, based on the exclusion’s criteria it would be useful if the authors clarify why

Changed

Lines 176

one of the studies selected refers to children and adults

Figure 2 is not properly explained, and in any case, it

would be more appropriate if moved to line 161

Changed

Lines 264-273

Section 3.4 could be revisited better organising the content, moving figure 3 down, after a brief explanation of what is the meaning of the risk of bias, Concurrently the comments related to the different authors, lines 180-205, should be presented in a better and more organic form, in relation to the section (no bold style for

comment lines 204-205)

Changed

Lines 314-348

Conclusion could be expanded giving more attention to the fact that children suffer racial disparities as a result of the racial disparities and overall discrimination suffered by their parents hence the racial gap in disease is a “by-product” of the parents’ racial socio-economic disparities

Changed

Lines 446-455

Thanks for your feedback!

Round 2

Reviewer 2 Report

accept